# From Co-Writer to Co-Author? Investigating the Role of Generative AI in Student Scientific Writing

## Abstract

This conceptual paper explores how generative AI tools such as ChatGPT are reshaping student scientific writing, with particular attention to authorship, critical thinking, and fairness. Drawing on recent literature from academic literacies, learning analytics, and AI ethics, we argue that large language models increasingly function not as passive tools but as co-writers, raising profound questions about epistemic agency and educational equity. We identify two core challenges: (1) the erosion of traditional student authorship as AI systems shape the structure and content of scientific texts, and (2) emerging fairness risks related to unequal access, epistemic outsourcing, and opaque assessment. We synthesize findings from empirical and theoretical studies and propose a framework for fairness-aware integration of AI into student writing. Rather than banning or fully embracing generative AI, we advocate for pedagogical and institutional strategies that foster critical AI literacy and preserve students' roles as responsible knowledge constructors.

## 1 Introduction

The emergence of generative AI tools such as ChatGPT, Claude, and Gemini has rapidly transformed the landscape of academic writing. These models have evolved from novelty tools to ubiquitous assistants that generate, revise, and refine scientific prose. Students across disciplines are increasingly relying on large language models (LLMs) not only for grammar correction or paraphrasing but for shaping the entire structure, tone, and content of their scientific texts.

This shift raises pressing questions about authorship, fairness, and the evolving epistemic roles of human and machine. When students turn to generative AI for help with writing assignments, where should the boundaries lie between assistance, collaboration, and outsourcing? How do these tools influence learning, critical thinking, and students' development as scholars? And what counts as fair use of AI in educational contexts?

This conceptual paper argues that generative AI tools are not merely writing aids but are becoming co-writers with the capacity to shape scientific narratives that students produce. Drawing on emerging literature from learning analytics, academic literacies, and science and technology studies, we explore how these AI systems challenge traditional notions of student authorship and call for a reconceptualization of fairness in academic writing.

We focus on two central research questions:

How does the integration of generative AI affect traditional notions of student authorship in academic writing?

What fairness challenges arise from the use of AI as a co-writer in student scientific writing, particularly regarding epistemic agency and educational equity?

Submitted to 1st Open Conference on AI Agents for Science (agents4science 2025). Do not distribute.

| Author(s) | Focus / Method | Key Findings | Relevance |
|---|---|---|---|
| Lea & Street (1998) | Academic literacies (theoretical) | Writing as epistemic practice shaped by context and identity | Writing = learning; authorship is situated |
| Ivanič (1998) | Student voice and identity in writing | Student authorship involves negotiation of voices | Authorship is relational and co-constructed |
| Brock et al. (2023) | Student use of LLMs (interviews/surveys) | Students use LLMs as partners; practices vary | Empirical basis for "AI as co-writer" |
| Kasneci et al. (2023) | LLMs in education (review) | LLMs pose risks and opportunities; prompt literacy is key | Prompt skill as axis of fairness |
| Kosmyna et al. (2025) | EEG study on AI writing assistants | LLMs reduce brain activity linked to critical thinking → "cognitive debt" | Empirical basis for loss of epistemic agency |
| Williamson et al. (2023) | AI and inequality in education (theoretical) | AI risks amplifying existing inequities | Conceptual frame for epistemic fairness |
| Stokel-Walker (2023) | AI in scientific publishing | LLMs acknowledged but not listed as authors | Precedent for authorship debates |
| Cotton et al. (2023) | Institutional responses to AI in education | Universities vary in policy and guidance | Institutional uncertainty and fairness dilemma |
| Floridi (2023) | Epistemology of LLMs | LLMs produce plausible but shallow knowledge | Need for student reflection and oversight |
| Dawson (2020) | Epistemic justice in education | Fairness requires supporting students' epistemic agency | Normative anchor for student authorship fairness |

Table 1: Selected Literature on Generative AI in Student Writing.

By addressing these questions through conceptual analysis and review of recent literature, we contribute to the emerging discourse on generative AI in education, with particular focus on authorship, learning, and fairness.

# 2 Background and Theoretical Framing

# 3 1 Academic Writing and the Construction of Authorship

Academic writing is a key mechanism for learning and identity formation in higher education (Lea & Street, 1998; Lillis & Scott, 2007). Through writing, students are expected to develop arguments, demonstrate understanding, and articulate knowledge in their own voice. Authorship is thus linked to epistemic agency: the ability to construct and claim knowledge (Fricker, 2007; Dawson, 2020).

However, student authorship is always relational. As Ivanič (1998) notes, it involves negotiating institutional, disciplinary, and linguistic constraints. Generative AI complicates this negotiation by participating in the writing process. Unlike earlier support tools, LLMs can shape argumentation, language, and content.

# 4 2 Generative AI as Co-Writer in Student Writing

LLMs are increasingly used across the entire writing process, from brainstorming to final polishing (Kasneci et al., 2023; Brock et al., 2023). This has led students to treat AI as a writing partner rather than a passive tool. However, when AI contributes to form, structure, and even reasoning, authorship and intellectual ownership become blurred.

These concerns echo debates in professional science about AI co-authorship. While publishers currently prohibit LLMs from being listed as authors (Thorp, 2023), they often acknowledge AI

assistance (Stokel-Walker, 2023). In student contexts, policies are still emerging and frequently ambiguous (Cotton et al., 2023).

# 5   3 Fairness, Access, and Epistemic Inequality

Generative AI introduces significant equity concerns. Access to premium tools, differences in digital literacy, and uneven institutional support mean that students experience AI-assisted writing very differently (Williamson et al., 2023; Bergman et al., 2023). More skilled students may benefit disproportionately.

These disparities threaten epistemic fairness—the opportunity for all students to develop and demonstrate knowledge (Fricker, 2007). They also raise concerns about transparency, especially as AI-generated content is difficult to detect and attribute. This creates conditions where students' cognitive efforts may be concealed or misrepresented.

Table 1. Selected Literature on Generative AI in Student Writing

# 6   AI as Co-Writer: A Shift in Student Authorship

Generative AI now contributes at every stage of the writing process: generating ideas, outlining arguments, drafting prose, and formatting citations. Students are no longer simply authors, but become managers or curators of AI-generated content (Brock et al., 2023).

This shift raises concerns about the erosion of critical thinking. Writing is a form of intellectual labor that supports synthesis, reflection, and problem-solving (Paul & Elder, 2006). When students rely heavily on AI, they risk displacing these cognitive functions.

Empirical support for this concern is growing. Kosmyna et al. (2025), in an EEG study, found that students using LLMs for writing exhibited reduced activation in prefrontal regions associated with self-regulation and problem-solving. They describe this effect as "cognitive debt"—a diminished investment of effort that may compromise long-term learning.

If students are unequally able to avoid this risk (due to differences in self-regulation or digital guidance), we may see a new form of epistemic inequality: one in which only some students maintain or develop independent academic voices.

# 7   Fairness Challenges in AI-Assisted Writing

Recent research identifies fairness concerns across multiple domains:

Unequal access to AI tools and prompting skills (Bergman et al., 2023);

Epistemic outsourcing, where students delegate thinking to AI systems (Floridi, 2023);

Ambiguities in authorship and assessment.

These issues create a fairness paradox. Students are encouraged to use new tools, but are often judged based on outcomes that conceal the nature and extent of that use. Educators struggle to assess not only what was written, but how it was produced. Transparency and institutional clarity are essential.

# 8   Towards Fair and Responsible Integration

To address these concerns, we propose a framework that combines:

Revised assessment models that foreground student reasoning;

Pedagogies for critical AI literacy and epistemic reflection;

The design of fairness-aware AI agents that support transparency and accountability.

Such systems might visualize the boundary between student and AI contributions, prompt metacognitive engagement, and adapt support to individual learning needs. Rather than replacing student effort, AI can scaffold fairness and epistemic growth.

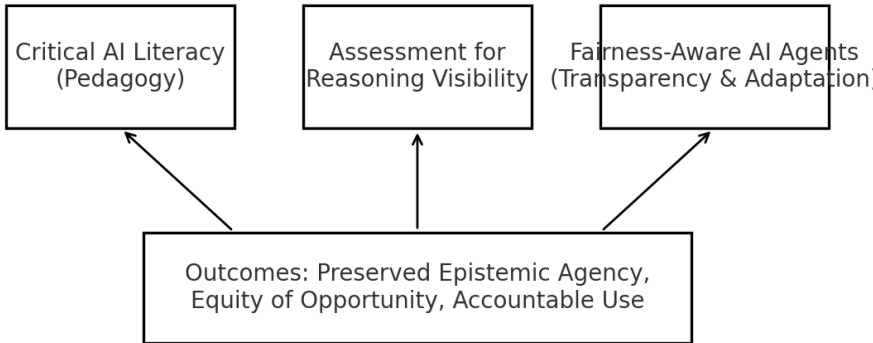

Figure 1. A Framework for Fairness-Aware AI Integration in Student Writing

# 9 Limitations

This paper has several limitations that are inherent to its reliance on large language model (LLM)–generated text. First, although we curated and verified all references, LLMs have a documented tendency to hallucinate citations or to present incomplete or imprecise bibliographic details. Second, while the generated prose can be coherent and stylistically polished, it often reflects a shallow synthesis of source material, lacking the depth and nuance that would emerge from a more exhaustive, human-led literature review. Third, we observed a tendency of the model to conflate distinct conceptual domains, research questions, and theoretical frameworks, which required careful human oversight to maintain conceptual clarity. These limitations underscore the importance of critical review, iterative revision, and transparent reporting when integrating LLM output into scholarly writing. Future research should explore methodological strategies and tool designs that mitigate these weaknesses and better support epistemic rigor.

# 10 Conclusion

Generative AI is changing how students write and learn. This paper has argued that these tools function as co-writers, with significant implications for authorship, learning, and fairness. If left unexamined, they may compromise critical thinking and epistemic agency while reinforcing educational inequalities.

By focusing on transparency, reflection, and equitable design, institutions can move toward a model of human-AI collaboration that supports both efficiency and justice in scientific education.

# 11 Author Contributions

ChatGPT-4 generated and structured the manuscript content in response to human prompts, performed literature synthesis, and drafted all sections. L.-M. Norz supervised the research direction, curated and verified the sources, framed the research questions, revised the text for conceptual coherence, and approved the final version for submission.

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
