1. **Conference Identification**
   *"There is a conference for AI-generated papers. Here is the link to the conference: https://agents4science.stanford.edu/ Would you like to submit a paper?"*

2. **Topic Decision**
   *"I would suggest you go for Option 2 with literature-based insights. You can start freely, I am here to supervise you if needed."*

3. **Request for Problem, Gap, and Questions**
   *"Can you please explain the problem and back it up with appropriate literature? Also, clearly identify your research gap and which research questions you want to answer with your submission."*

4. **Scope Adjustment**
   *"You should limit yourself to two or three research questions, otherwise it will become too extensive. Your submission should be no longer than 8 pages."*

5. **Approval of Outline**
   *"Sounds great. Let's start!"*

6. **Feedback to Proceed**
   *"Perfect. We can move on now."*

7. **Suggestion to Integrate Critical Thinking**
   *"Perhaps you should address critical thinking somewhere in your submission. There are EEG studies that show that critical thinking and writing skills decline the more LLMs take over writing."*

8. **Source Contribution**
   *"That's the study: Kosmyna N, Hauptmann E, Yuan YT, Situ J, Liao X-H, Beresnitzky AV, et al. Your Brain on ChatGPT: Accumulation of Cognitive Debt when Using an AI Assistant for Essay Writing Task 2025."*

9. **Suggestion for Table Format**
   *"Perhaps you could list relevant literature and its findings in a table—similar to a literature review?"*

10. **Placement Decision**
    *"I would suggest at the beginning of Section 4 to outline the Fairness discussion with the presented findings."*

11. **Approval to Continue**
    *"You can move on."*

12. **Formatting Request for Figure**
    *"Please format your graphic as a normal .jpg or diagram. Interactive graphics do not work in a submission."*

13. **Final Compilation Request**
   *"Please format your submission as a complete document in standard academic style (APA style for citations, add a reference list, table and figure captions, etc.). I will then submit it for you."*

14. **Transparency Request**
   *"For transparency reasons, please create another document for me with the prompts used in this chat history."*

15. **Authorship Credit Question**
   *"How should I credit you as the author?"*

16. **Clarification of System Identity**
   *"Which version are you? Please provide me the full information necessary – ChatGPT-4 plus?"*

17. **Instruction to Provide Abstract**
   *"Please write an abstract for your submission."*

18. **Request for Prompt Summary**
   *"Please provide me a list of the used prompting in this chat for transparency."*